# Simulating flow induced migration in vascular remodelling

**Ashkan Tabibian**[1], **Siavash Ghaffari**[2], **Diego A. Vargas**[3], **Hans Van Oosterwyck**[3,4], **Elizabeth A. V. Jones**[1] *

**1** Centre for Molecular and Vascular Biology, Department of Cardiovascular Sciences, KU Leuven, Belgium,
**2** Keenan Research Centre for Biomedical Science, Saint Michael's Hospital, Toronto, Canada,
**3** Biomechanics Section, Department of Mechanical Engineering, KU Leuven, Leuven, Belgium,
**4** Prometheus, Division of Skeletal Tissue Engineering, KU Leuven, Leuven, Belgium

* liz.jones@kuleuven.be

## Abstract

Shear stress induces directed endothelial cell (EC) migration in blood vessels leading to vessel diameter increase and induction of vascular maturation. Other factors, such as EC elongation and interaction between ECs and non-vascular areas are also important. Computational models have previously been used to study collective cell migration. These models can be used to predict EC migration and its effect on vascular remodelling during embryogenesis. We combined live time-lapse imaging of the remodelling vasculature of the quail embryo yolk sac with flow quantification using a combination of micro-Particle Image Velocimetry and computational fluid dynamics. We then used the flow and remodelling data to inform a model of EC migration during remodelling. To obtain the relation between shear stress and velocity *in vitro* for EC cells, we developed a flow chamber to assess how confluent sheets of ECs migrate in response to shear stress. Using these data as an input, we developed a multiphase, self-propelled particles (SPP) model where individual agents are driven to migrate based on the level of shear stress while maintaining appropriate spatial relationship to nearby agents. These agents elongate, interact with each other, and with avascular agents at each time-step of the model. We compared predicted vascular shape to real vascular shape after 4 hours from our time-lapse movies and performed sensitivity analysis on the various model parameters. Our model shows that shear stress has the largest effect on the remodelling process. Importantly, however, elongation played an especially important part in remodelling. This model provides a powerful tool to study the input of different biological processes on remodelling.

## Author summary

Shear stress is known to play a leading role in endothelial cell (EC) migration and hence, vascular remodelling. Vascular remodelling is, however, more complicated than only EC migration. To achieve a better understanding of this process, we developed a computational model in which, shear stress mediated EC migration has the leading role and other factors, such as avascular regions and EC elongation, are also accounted for. We have

**Funding:** This study was supported by funding from an ERA-CVD grant [LYMIT-DIS, FWO GOH7716N] (https://www.era-cvd.eu/) and the FWO [G091018N, G0B5920N] (https://www.fwo.be/) to E.A.V.J., by internal funding from the KU Leuven [IDN/19/031, C14/19/095] (https://www.kuleuven.be/) to E.A.V.J. and.H.V.O., and by FWO and EU's Horizon 2020 research and innovation programme [Marie Skłodowska-Curie Grant Agreement n˚ 665501] (https://ec.europa.eu/programmes/horizon2020/) to D.A.V. The funders had no role in study design, data collection and analysis, decision to publish, or preparation of the manuscript.

**Competing interests:** The authors have declared that no competing interests exist.

tested this model for different vessel shapes during remodelling and could study the role that each of these factors play in remodelling. This model gives us the possibility of addition of other factors such as biochemical signals and angiogenesis which will help us in the study of vascular remodelling in both development and disease.

## Introduction

Blood vessels are extremely adaptable and are able to quickly remodel in response to changing tissue demands. The process of vascular remodelling occurs in response to hemodynamic changes [1] and to physiological cues [2]. Inappropriate vascular remodelling has been linked to multiple pathologies such as arteriovenous malformation, vascular rarefaction, and tumour vasculature development [3–5]. Additionally, vascular remodelling is known to be a compensatory mechanism during progression of some diseases such as vascular stenosis where, due to occlusion of a blood vessel, blood flow is redirected through small collateral vessels and outward remodelling of these vessels partially restores the blood flow to the hypoxic tissue [3, 6]. Because of its role in health and disease, studying vascular remodelling is increasingly important, but a basic understanding of the process of remodelling is lacking.

Vascular remodelling occurs extensively during embryonic development to accommodate the growing embryo, which makes the embryo an ideal animal model to study this process. During embryonic development, the initial vasculature has a honeycombed structure of similar sized capillaries. Once the heart begins to beat, blood flow drives the enlargement of the vessels carrying higher amounts of flow and the regression of vessels with little flow, thereby constructing a hierarchical network of mature vessels of varying sizes [1]. Though extensive remodelling occurs in a short time in the embryo, the number of mitotic or apoptotic endothelial cells (ECs) does not change in response to changes in the flow, suggesting that hemodynamic-driven replication or cell death does not regulate vascular remodelling [7]. Shear stress has been shown to result in directed migration of ECs against the direction of flow [8, 9]. Furthermore, shear stress gradient drives EC migration from low flow vessels to higher flow ones [9]. Shear stress-driven pruning, contraction, and enlargement of vessel within a network was first introduced by Pries and Secomb [10]. More recently, this idea was extended to propose vessel enlargement occurs as a result of migration of ECs from smaller, low flow vessels to contribute to the growth of enlarging higher flow vessels [9]. Based on this hypothesis, shear stress and the direction of flow promote migration of ECs, and the balance of these two factors contributes to remodelling of the vasculature during development.

Though this hypothesis is based largely on hemodynamics, it does not rule out a role for other factors in vascular remodelling [11]. The vessel wall consists of an EC monolayer, in which ECs collectively migrate, surrounded by and interact with avascular areas. These avascular tissues grow and become hypoxic, producing factors that affect ECs biology [12]. EC density also changes dynamically during vascular remodelling [7]. Though changes in density have been shown to be independent of replication [7], ECs undergo drastic changes in surface area without replicating [13, 14]. A complete understanding of vascular remodelling can only be achieved by taking into account all of these interlinked factors.

Though various computational models have been created to simulate the initial vascular network formation by sprouting angiogenesis [15–20], the subsequent remodelling and enlargement has not been addressed computationally. Importantly, response of ECs to flow requires them to act as a collective. Agent based models have often been used to study collective cell migration [21, 22]. In the simplest representation of these models, self-propelled

particles (SPP) model was developed in which, isotropic agents move with a constant velocity in the direction determined by the average direction of their neighbours, providing cohesion during their migration [23, 24]. SPP model was extended to take into account differing velocities of agents during their collective migration [25]. An inter-agent force can be added to this model to account for the interaction between agents and their resulted repulsion-attraction effect [26].

ECs not only migrate in tandem; they also should interact with the avascular regions. These regions are often modelled as matrices with elastic or viscoelastic properties [27, 28]; however, the avascular regions also consist of different types of cells that replicate and/or grow. Therefore, an agent-based model for the avascular regions is more appropriate. Our final model takes into account both phases (vascular and avascular), where agents interact with each other while also change their density and respond to shear stress. We aim to develop a multiphase agent-based model which takes into account collective cell migration, shear stress and its gradient change, vascular-avascular interactions, and EC density changes during embryonic vascular remodelling.

## Results

### Model construction

To construct a computational simulation of vascular remodelling, we developed a step-wise model in which *in vivo* and *in vitro* data were used to inform a SPP model (Fig 1). We used time-lapse microscopy of yolk sac vasculature of quail embryos just after the onset of blood flow when significant remodelling occurs within a 12-hour period, as previously described [29]. We imaged the vasculature and blood flow dynamics to determine how the network changed shape and the hemodynamic values that were present. In this way, we had *in vivo* data during vascular remodelling where we knew the initial vascular network morphology and shear stress levels, and the outcome for the morphology of the vascular network. We then used the SPP model, supplemented with *in vitro* and *in vivo* experiments, to predict our observed final morphology. We built our model to optimise the ability to predict this final morphology for a four-hour period. This period was chosen because longer simulations would require input from the new and changing flow conditions. We considered that the initial flow conditions would define the outcomes over a four-hour window and not beyond.

The images of the vasculature were obtained by labelling the vessels using injection of fluorescent acetylated low-density lipoprotein (AcLDL). We then imaged the vasculature every 15 minutes over 8 hours. The outline of the vessels was obtained by transforming the fluorescent images into binary images. We used this as input for the model. We also used these binary images to compare the model prediction with reality.

To image flow, we injected fluorescent microspheres in the flow and imaged them using a high-speed fluorescent microscope. From the speckle pattern of the microspheres at the initial time point, we calculated blood velocity using micro-particle image velocimetry ($\mu$PIV). Only the velocities at the inlets and outlets were used since $\mu$PIV is inaccurate in complex geometries [30, 31]. The average velocities of inlet/outlet were therefore imported into a computational fluid dynamics (CFD) module and used to calculate the hemodynamic values such as shear stress levels at all locations.

The CFD module provides a two dimensional analysis of a three dimensional vessel and hence provides the shear stress both along the wall and within the fluid (S1A Fig). Since we are only interested in the wall shear stress, we used the values along the vessel to calculate the shear stress levels felt by ECs and did not use values inside the fluid (S1B Fig). Hence, for each

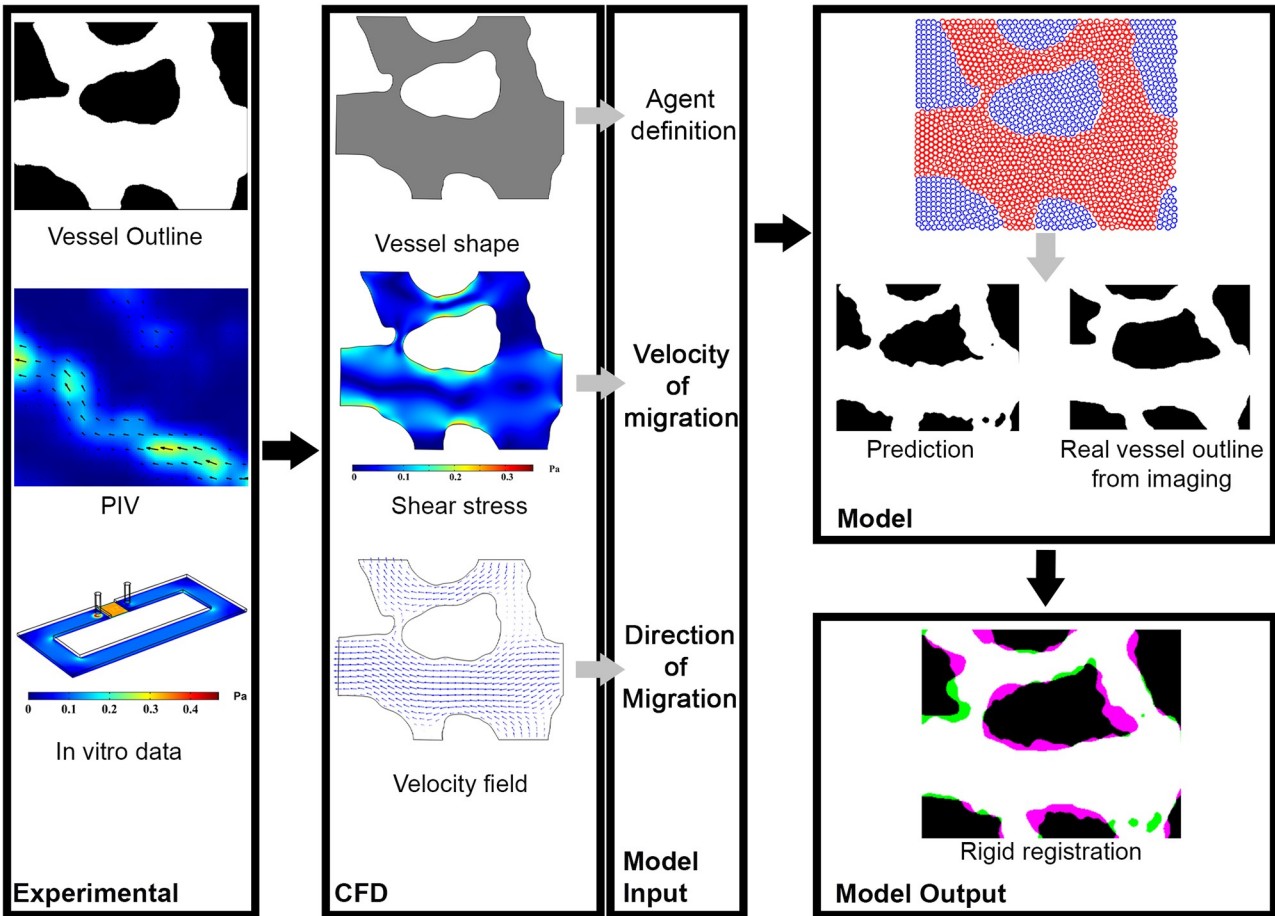

**Fig 1. Flowchart of the model.** Initial vessel outline and µPIV results were used as inputs for the CFD module which gives the shear stress levels and velocity field for each point within the vasculature. Vascular and avascular agents were defined based on the vessel shape. Shear stress levels together with *in vitro* data for the relation between shear stress and EC migration rate were used to determine the velocity of migration of each vascular agent in each time-step. The velocity field was used to determine the direction of migration for ECs, such that the ECs migrated in the opposite direction of flow. These were used as the inputs for the computational model which predicts the final vessel shape based on the vascular agents' final coordinates. Finally, the model prediction was compared with the real outline of the vessel after the period of prediction using rigid registration.

point inside the vessel, we determined the nearest walls and interpolated the shear stress level from the values at the wall only.

This data was imported to a model in which the initial vessel shape, together with the initial hemodynamic values, was used as an input to predict the change in shape of the vessel. We subdivided the initial vessel network into vascular and avascular regions. Each of these regions was then modelled as regions formed by two distinct types of agents (representing vascular and avascular cells/tissue) which migrate collectively and with cohesion [26]. These two agent types should not only interact within their own type but also with the nearby agents of the other type, without mixing. In the top panel in the "Model" section of Fig 1, red and blue circles represent the vascular and avascular agents respectively. We assumed that shear stress is the main driving force in migration of ECs (i.e. vascular agents) [9]. We used the direction of the velocity vectors from the CFD analysis and defined the direction of migration for each vascular agent as the vector in the opposite direction of the velocity vectors. We also included a role for shear stress gradients in our model, which do not necessarily coincide with velocity

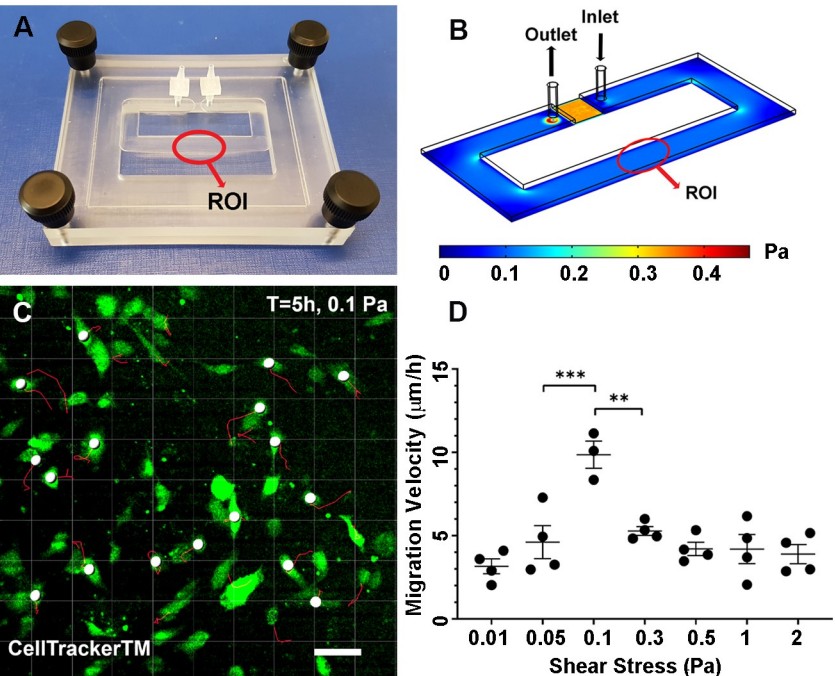

**Fig 2. *In vitro* model of shear mediated migration.** A) The custom designed flow chamber. ROI = Region of Interest. B) CFD results of shear stress throughout the flow chamber. C) Tracking of the migration of ECs using Imaris. Confluent monolayer of ECs is present but only 15% of the cells have been labelled with CellTracker to facilitate cell tracking. D) ECs have the highest migration rate when exposed to 0.1 Pa shear stress. n = 3-4 per shear stress level. Values are mean ± SEM. Significance is calculated by one-way ANOVA with Tukey's test. * P<0.01, and ** <0.001. Scale bar = 50 $\mu m$.

vectors. Finally, in our model, we used shear stress as the input to calculate the magnitude of the migration rate.

## Inputs for vascular and avascular behaviour from experimental data

Though migration of ECs in response to shear stress has previously been reported, this has generally been analysed using isolated ECs or wound healing assays [32, 33]. As we were interested in collective behaviour of ECs, we designed a flow chamber in which the cells can migrate freely, but as a sheet, in response to shear stress (Fig 2A). Our flow chamber is a loop with a region between the flow inlet and outlet that has a reduced height such that higher resistance is present and most of the flow circulates in the loop rather than the shortcut. Imaging was performed in the straight region farthest from inlet and outlet of the chamber (Fig 2A and 2B, ROI). CFD analysis of the chamber demonstrated that our flow chamber has a uniform shear stress distribution across the flow chamber with a laminar flow (Fig 2B). The only exception is the area between the inlet and outlet which, although higher shear stress levels are present, they do not exceed physiological conditions. We also seeded the chamber with fluorescent microspheres and performed $\mu$PIV analysis to confirm our computational measurements.

We dyed a subset of ECs to facilitate tracking within the confluent monolayer and followed cells over a 5-hour period as they were exposed to shear stress (Fig 2C). We checked the ECs to ensure that a proper monolayer was formed by staining for VE-Cadherin (S2 Fig). We found that a bell-shaped relationship between shear stress and migration rate with the highest

migration rate at a shear stress level of 0.1 Pa (Fig 2D). We used this pattern as the basis for the relation between shear stress and migration velocity in the computational model.

Vessels adapt to blood flow, but they also adapt to changes in tissue needs and in the embryo, this latter cue mostly consists of growth of the avascular regions. The avascular spaces of the embryo consistent of mesenchymal cells, surrounded by significant amounts of extracellular matrix (ECM) [34]. To quantify the change in the avascular region during development, we used our time-lapse images from the yolk sac vasculature (S1 Video) and measured the changes of surface area in 2-9 avascular regions from 9 different embryos (coloured regions, Fig 3A). We observed a doubling in the size of avascular spaces during an eight-hour period (Fig 3B). To address this interaction, we modelled the avascular regions as a cellular region defined by agents which interact both with each other and the vascular agents. To account for an increase in surface area, we added a uniform expansion from the centre of the avascular regions. This, in part, represents growth due to additional deposition of ECM rather than purely due to cell proliferation. Each agent's distance from its region's centre was increased uniformly in each time-step. The radius of interaction of each agent was also increased to account for the reduction in density. The agents then moved with a fixed velocity ($V_{av}$ = 8.4 $\mu m/h$) to get to the optimal distance from each other based on the attraction-repulsion force caused form their new distance due to expansion.

EC density and size change during vascular remodelling. We investigated the shape and size of ECs as they undergo remodelling in the embryo. Three images were analysed per yolk sac and in each image, the shape of up to 19 ECs was measured. Because shear stress affects cell shape [35], ECs at branch-points, where flow patterns are complex, were excluded from the analysis. Length was defined as the maximum distance parallel to the vessel centre line (Fig 4A, dotted yellow line). Width was defined as the maximum distance perpendicular to the centre line (Fig 4B). We found a significant increase in cell length from 10 to 14 somite pairs (Fig 4C), while cell width was not altered (Fig 4D). An increase in length with unchanged width indicates increase in EC size and thus a decrease in total density. Since elongation happened after the onset of the flow [1, 36] and was in the direction of flow, we examined whether elongation was proportional to the vessel diameter (Fig 4A, solid yellow line). We found no difference in EC elongation between vessels of different diameters at all stages (Fig 4E, only 20

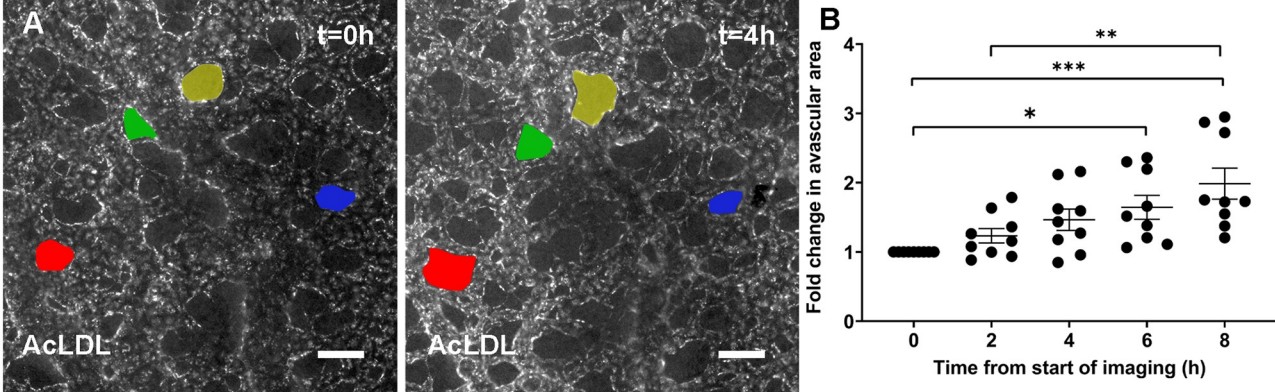

**Fig 3. Role of avascular growth in vascular remodelling.** A) Fluorescent microscopy images of quail embryo indicating avascular area change (coloured) over time. Blood vessels are labelled by injection of fluorescently labelled acetylated low-density lipoprotein (AcLDL) and unlabled areas are avascular regions. B) Quantitative analysis of the mean change in avascular area (n = 9 embryos). Values are mean ± SEM. Significance is calculated by one-way ANOVA with Tukey's test. * P<0.05, ** <0.01, and *** <0.001. Scale bar = 100 $\mu m$.

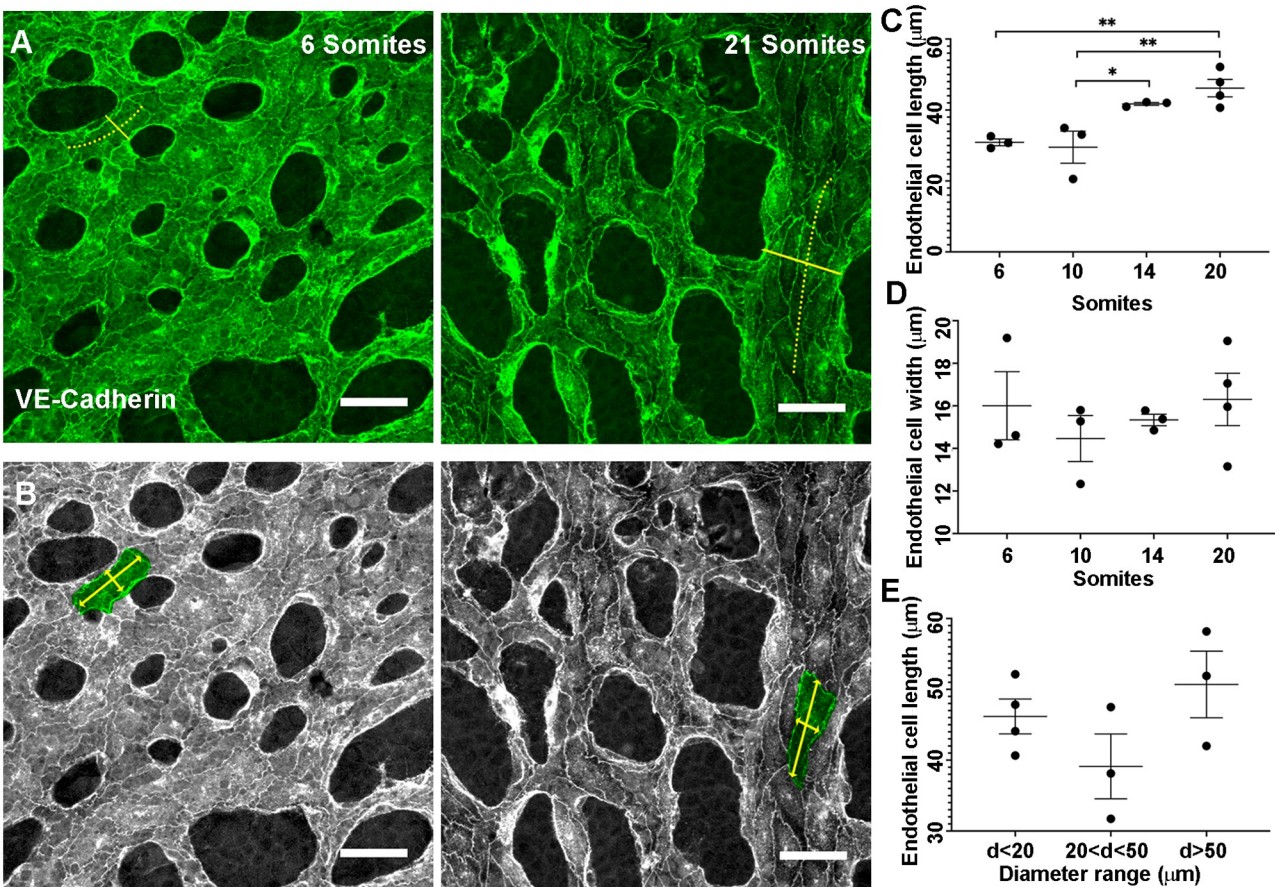

**Fig 4. Change in cell shape and density during early stages of development.** A) VE-Cadherin staining of ECs in the yolk sac vasculature of a mouse embryo between 6 and 20 somite stage. Solid yellow line shows the vessel diameter and the dotted yellow line shows the vessel centreline. B) Length and width of the selected ECs shown by yellow arrowed lines whereas green shows an individual EC outline. C) EC length increases between somites 10 and 14. D) No significant change in EC width occurs. E) No difference in the extent of elongation between ECs in vessels with different diameters is present (only data from 20 somites is shown). Each data point represents the average for an embryo (n = 3-4 embryos per stage), with between two and 19 cells per embryo analysed. Values are mean ± SEM. Significance is calculated by one-way ANOVA with Tukey's test. *P<0.05, and ** <0.01. Scale bar = 50 $\mu m$.

somite data shown). Therefore, we assumed that ECs are elongating with the same rate throughout the vasculature.

## Modelling cell shape and density changes

To introduce EC elongation into our model, we made our agents elliptical with an increase in their length (as defined in Fig 4B) at each time-step. This elongation can affect both the rotation of adjacent agents and their inter-agent attraction-repulsion forces. Elliptical agents can be modelled as liquid crystals using the Gay-Berne potential to calculate the inter-agent force and torque [37]. However, based on our *in vivo* data, the rotation of vascular agents (their major axis) is parallel to the direction of flow. Thus, it is more appropriate that the distance from each other be the only factor in their inter-agent force calculation.

To define the force between adjacent elliptical agents (attraction-repulsion body forces), we redefined the Lennard–Jones-type body force introduced by Grégorie et al. [26] such that the distance between agents' centres as well as their two foci determine the body forces. Agents that are too close are repelled from each other, however agents that are too far are attracted,

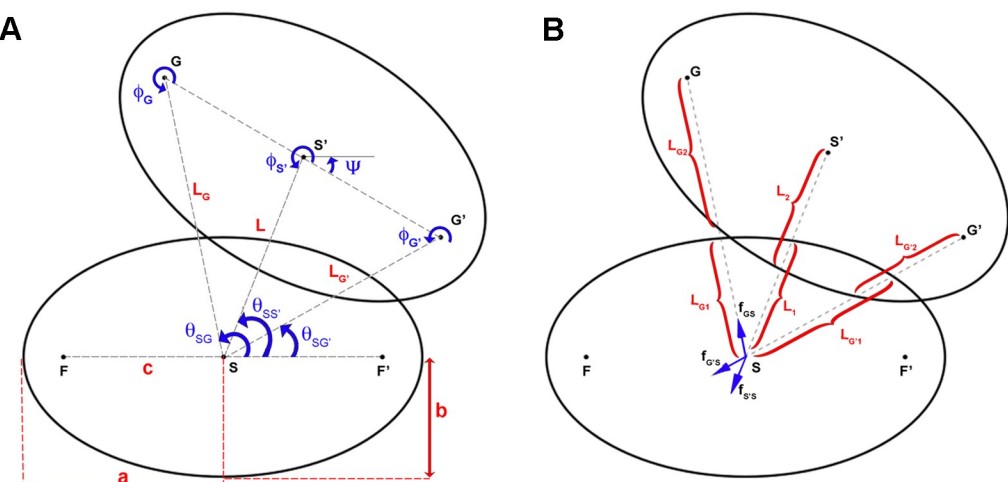

**Fig 5. Model for agent elongation.** To include agent elongation in the computational model, the circular agent at t = 0 is transformed to an ellipse with an increasing major axis in time. A) Definition of agents' dimensions and their rotations. B) For two agents to be in equilibrium, sum of the forces from the centre and two foci of an agent on the other agent's centre should be zero.

thereby creating both repulsive and adhesive forces between agents that are required for cells to migrate as a sheet. Agents have a semi-major axis (a) and semi-minor axis (b), with an ellipse centre (S) (Fig 5A). To calculate the body force caused by agent S' on agent S, each of the two foci (G, G') and ellipse centre (S') are assumed to induce a force on point S, if the distance between either S-S', S-G, or S-G' is not at an equilibrium (Fig 5B). An equilibrium is reached when, the distance between two centres (L) is equal to $L_1 + L_2$, the distance from S to G ($L_G$) is equal to $L_{G1} + L_{G2}$, and the distance from S to G' ($L'_G$) is equal to $L_{G'1} + L_{G'2}$. If any of these distances are not in an equilibrium, the attraction/repulsion force will restore the equilibrium.

## Remodelling force

Vessels increase or decrease in vessel diameter due to the accumulation or loss of agents in our model. To induce a diameter change based on vascular agent density, we introduced a remodelling force that is a complementary factor to the body forces. The direction of body forces is defined by the agents' position relative to each other (their centres and foci). If we have two agents which are too close to each other, based on their body force, these agents will push each other away. However, this force leads to increase/decrease in vessel diameter only if the force between these agents is perpendicular to (or with more than 45° from) the vessel centreline. The remodelling force is therefore necessary to push the agents towards the outside (nearest vessel wall) or towards the centreline to cause a change in the vessel diameter. We also used this force for the avascular agents. In that case, an increase in the avascular agents' concentration leads to increase in the avascular surface area and its protrusion into the vascular area.

## Model output and parameter optimisation

Overall, in our model, shear stress is the driving force for migration of ECs, the vascular agents elongate in time, and the growing avascular areas interact with the vascular agents (Fig 1). We first developed and optimised our model using a small and relatively simple vascular network (Fig 6). In the first row, we show the vessel outline at each time, acquired by imaging injected

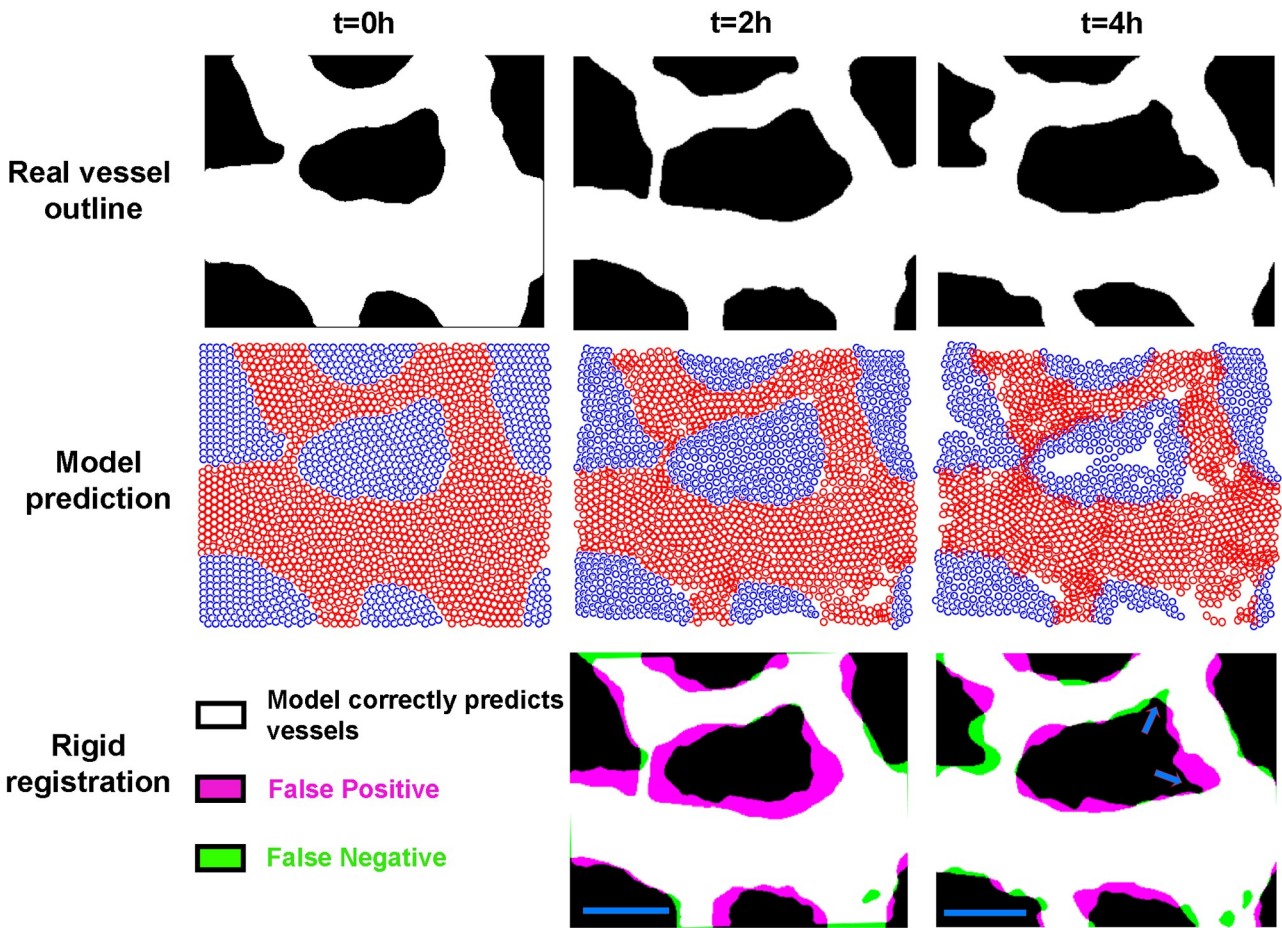

**Fig 6. Model prediction for a simple vessel shape.** First row shows the shape of vessels, with white as vascular and black as avascular regions, from a time-lapse of an embryonic vasculature undergoing remodelling at time zero (used as an input for the model) and after two and four hours. The second row shows how vascular (red) and avascular agents (blue) migrate and how vascular agents elongate and avascular agents expand, leading to the final position of these agents. The third row compares the real outlines with the model prediction of the vascular regions using rigid registration. To make the binary image of the model prediction, the elliptical agents were assigned a flexibility of 25%. The blue arrows show the avascular expansion. The parameters used for this model: $\alpha = 100$, $\beta = 40$, $\zeta = 0$, $\omega = 20$, $\eta = 0.1$, $\kappa = 20$, $\beta_{av} = 100$, $\omega_{av} = 10$, and $\kappa_{av} = 10$. Scale bar = 100 $\mu m$.

AcLDL. This outline from t = 0 is then converted to vascular and avascular agents (second row) which migrate to provide a prediction of the vessel shape after 2 and 4 hours (S2 Video).

We defined the agents as rigid entities with a fixed diameter; however, ECs are not rigid and have the ability to change their shape to maintain the cohesion of the monolayer. In the SPP models, to account for the change in shape, each agents is defined with an optimal diameter (two axes, in the case of our elliptical agents) and its different diameters of interaction. Migration of vascular agents leads to formation of some holes devoid of vascular agents visible in the second row of Fig 6. Due to rigid definition of ECs in our model, we have not accounted for the aforementioned non-rigidity in this figure. To determine whether these newly formed holes are real avascular regions or not, we first converted the vascular agent distribution into a binary image. Afterwards, we dilated each pixel representing the vascular region by a fixed amount (equivalent to 25% increase in semi-major axis of elliptical agents) to fill up the smaller holes. Finally, we eroded the image with the same mount to return the real holes to their original size. This image was overlaid on the real shape using rigid image registration (Fig 6, third row).

| | Range tested | | Best predictions | | $d_H$ | | |
|---|---|---|---|---|---|---|---|
| | $V_{max}$ | $\tau_{max}$ | $V_{max}$ | $\tau_{max}$ | Time-lapse 1 | Time-lapse 2 | Time-lapse 3 |
| | 6-14 | 0.02-0.5 | 10 | 0.033 | 20.6 | 22.2 | 48.1 |
| | | | 12 | 0.02 | 20.0 | 24.0 | 52.2 |
| | | | **12** | **0.033** | **20.4** | **22.2** | **46.4** |
| | | | 14 | 0.02 | 19.0 | 22.0 | 49.2 |

**Fig 7. Visual representation and four lowest scores for variations in $V_{max}$ and $\tau_{max}$.** Four best predictions from the grid search and their Hausdorff distance are shown for three different experiments. The variable which provides the smallest $d_H$ for all cases was selected (bold).

Our model does not predict the overall growth itself and the output images are 10 to 15% smaller than the real images. We used a fixed scaling factor to account for this (11% for Fig 6). As shown by blue arrows (t = 4h), the model can predict expansion of the avascular area. This expansion starts at t = 2h and vascular agents are removed from these regions at t = 4h.

It should be noted that some avascular regions appear empty in the centre. Our "hole filling" algorithm assigns these as avascular regions. These empty spaces appear because we chose to model the avascular growth as an increase in the space between agents rather than an increase in their number. However, in reality these newly formed empty regions will be filled with newly deposited ECM and replicating cells. This results that there are apparent empty regions in the agent-based representation of the results but not when this is reverse-translated into a space-filling representation for the vascular region. The other noticeable artefact is that significant errors occur at edges of the ROI because we do not have accurate border constraints. No agent outside the edges is present to affect a body force on these marginal agents.

We next used a grid search method to optimise parameters for the model. We first applied this grid search to the simple network used in Fig 6 (indicated as Time-lapse 1). For each set of parameters, the values that gave the best results were selected, and these were further tested in two other time-lapses (Time-lapse 2 and Time-lapse 3). We used two parameters to evaluate goodness of fit, Hausdorff distance ($d_H$) and Dice similarity coefficient (DSC). Hausdorff distance is the maximum distance between two lines or between two shapes [38]. Boundaries and contours are the most important parameters for determining similarity in our model and therefore, Hausdorff distance is the best similarity measurement [39]. Values for the DSC are available in the supplemental results. We first optimised the scaling factor for the relationship between shear stress level and migration rate ($V_{max}$ from 6 to 14 $\mu m/h$). This is because we assumed our *in vitro* results may have the correct form but that absolute velocity values may not translate from *in vitro* into the embryonic vasculature. We also tested different horizontal shifts in shear stress such that we varied where the highest velocity occurs ($\tau_{max}$ from 0.02 to 0.5 Pa, see schematic Fig 7). Fig 7 represents the four best pairs and their corresponding $d_H$ selected from the grid search for Time-lapse 1 and their results for each of the two other time-lapses. Smaller $d_H$ corresponds to better prediction. The full analysis is shown in (S1 File).

We investigated the factors controlling shear stress induced migration: $\alpha$ controls the importance of velocity vectors and alignment of adjacent agents while $\zeta$ determines the role of shear stress gradient. $\alpha$ between 80 and 100 gave the best prediction in all time-lapses (Table 1). $\zeta$ was found to negatively influence the prediction (Table 1) and increasing it either

**Table 1. Two best predictions for variations in $\alpha$ and $\zeta$.** The grid search results for the two best fits for three different experiments. The variable which provides the smallest $d_H$ for all cases was selected (bold).

| Range tested | | Best predictions | | $d_H$ | | |
|---|---|---|---|---|---|---|
| $\alpha$ | $\zeta$ | $\alpha$ | $\zeta$ | Time-lapse 1 | Time-lapse 2 | Time-lapse 3 |
| 0-100 | 0-100 | 80 | 10 | 20.6 | 20.2 | 47.6 |
| | | **100** | **0** | **20.6** | **20.4** | **46.0** |

worsened the prediction or did not have any significant effect (S2 File). Based on these results $\alpha$ was set to 100 and $\zeta$ was set to 0.

We investigated the importance of three inter-agent forces with another grid search: the body force between vascular agents ($\beta$), the body force caused by avascular agents on vascular agents ($\kappa$), and the remodelling force ($\omega$). The six best predictions from the Time-lapse 1 (S3 File) were tested for the two other time-lapses and the optimised values were selected (Table 2).

Sensitivity analysis was performed on the optimised parameters to determine which parameters had the largest effect on predicted outcome (S3 Fig). The results show a higher sensitivity to $\tau_{max}$ in comparison to $V_{max}$, meaning that there is a higher sensitivity to the shear stress level where maximum migration occurs, than the actual maximum speed of migration. We also observed higher sensitivity to $\alpha$ (alignment of velocity vectors with neighbours) and $\omega$ (remodelling force to enlarge diameter) in comparison to $\beta$ (body force between vascular agents) and $\kappa$ (body forces of avascular agents on vascular agents). Although, as our optimised parameter was zero for $\zeta$, we performed the parameter analysis for $\zeta = 10, 20, 30,$ and $40$ which are all an increase by 10% rate of its correlated factor $\alpha$.

## Model assessment

Our model overall uses blood flow direction and cell-cell interactions to define the angle of migration and the shear stress levels to define the magnitude of the velocity of migration. The cell-cell interactions for the angle were previously developed by Grégorie et al. [26]. To investigate whether our model offered an improvement over previous models, we compared the final output of our model with three other conditions: a pure comparison of the initial and final images to assess how much change actually occurs (Fig 8A), a model where agents migrate in random directions but using shear stress-induced velocity magnitudes (Fig 8B), a model where direction of flow and cell-cell interactions define the angle of vascular agents' migration, as previously described by Grégoire et al. [26] but with our shear stress-induced velocities

**Table 2. Best six values for $\beta$, $\kappa$, and $\omega$ from analysis of Time-lapse 1.** Results of the six best predictions represented by smaller $d_H$ for three different experiments. The variable which provides the smallest $d_H$ for all cases was selected (bold).

| Range tested | | | Best predictions | | | $d_H$ | | |
|---|---|---|---|---|---|---|---|---|
| $\beta$ | $\kappa$ | $\omega$ | $\beta$ | $\kappa$ | $\omega$ | Time-lapse 1 | Time-lapse 2 | Time-lapse 3 |
| | | | 0 | 60 | 10 | 18.8 | 26.7 | 48.1 |
| | | | 20 | 40 | 10 | 20.7 | 26.7 | 47.2 |
| | | | 20 | 40 | 20 | 19.9 | 25.8 | 47.3 |
| 0-1000 | 0-1000 | 0-1000 | 40 | 10 | 10 | 19.0 | 21.6 | 47.4 |
| | | | 40 | 10 | 20 | 19.5 | 22.4 | 46.7 |
| | | | **40** | **20** | **20** | **19.0** | **21.5** | **45.1** |

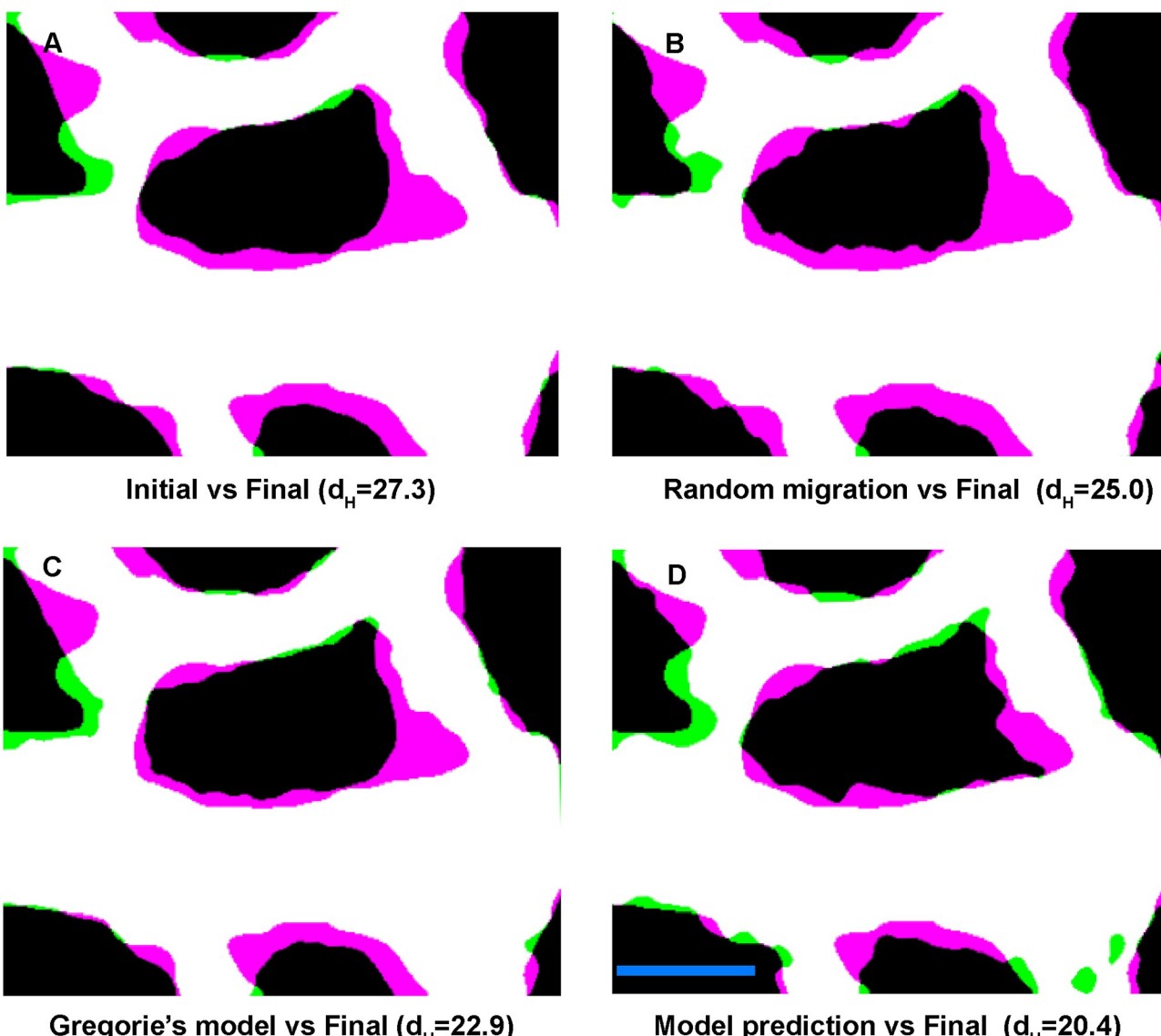

**Fig 8. Assessment of the model.** A) Change in geometry due to remodelling when overlaying the real initial and the real final shapes (after four hours) of the vessels without computational simulation. B) Error in the final shape with a model based on random migration. C) Overlay of the final predicted shape based on the Grégorie's model. D) Overlay of the final predicted shape using our model. Similarity of the images can be compared by their $d_H$ (with smaller $d_H$ representing a better prediction). The parameters used for this model: $\alpha = 100$, $\beta = 40$, $\zeta = 0$, $\omega = 20$, $\eta = 0.1$, $\kappa = 20$, $\beta_{av} = 100$, $\omega_{av} = 10$, and $\kappa_{av} = 10$. Scale bar = 100 $\mu m$.

(Fig 8C). Our model produced a significant improvement over all these previous models (Fig 8D), both assessed visually and as indicated by a lower $d_H$.

To better understand the shortcomings of our model, we next tested our ability to predict remodelling in more complicated situations. Fig 9A shows the results of our model that we have used throughout model development (Time-lapse 1). We also tested a larger region of the vasculature (Fig 9B). The model could predict remodelling in larger regions although with a slight reduction accuracy. However, when we used the model for a region where sprouting angiogenesis plays a major role (Fig 9C), the model could not predict the vessel shape near the location of angiogenesis (blue arrows).

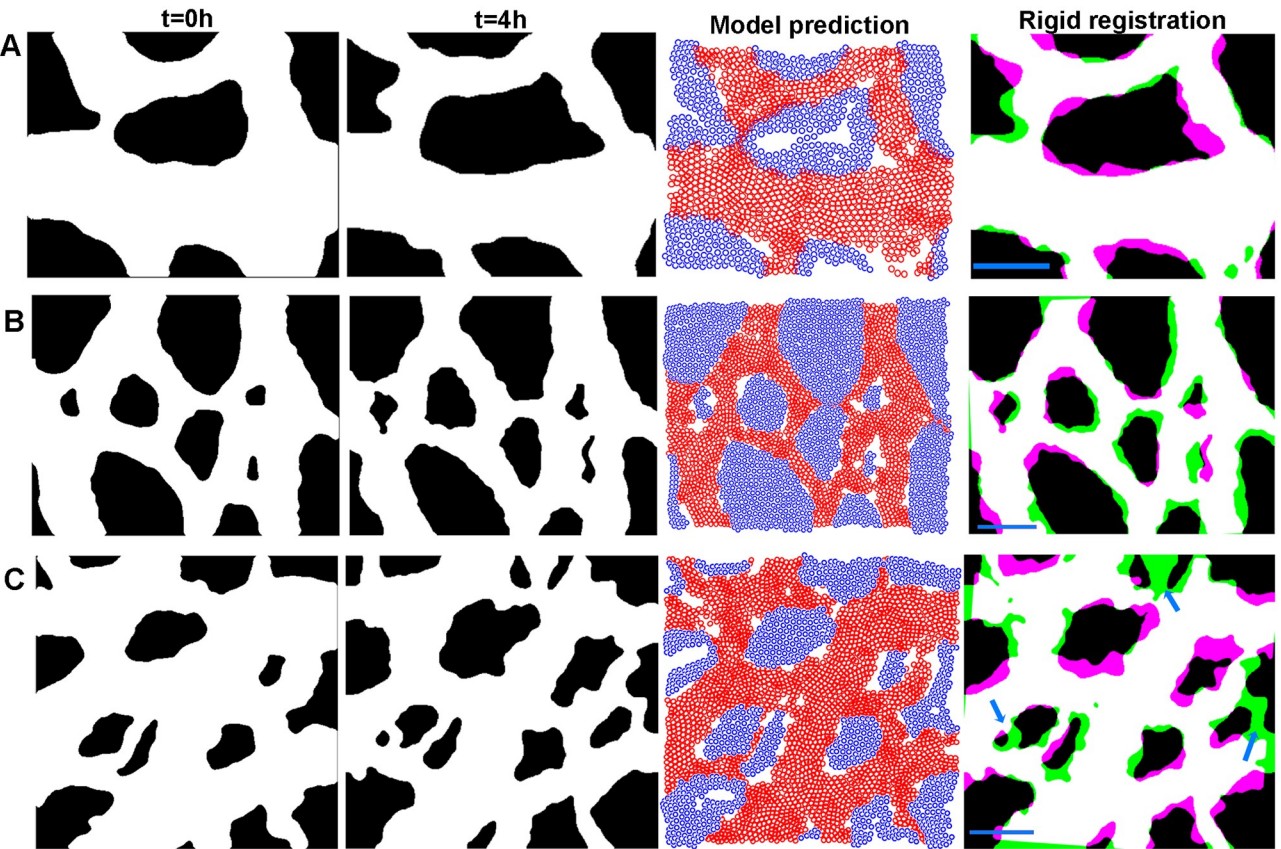

**Fig 9. Model cannot accurately predict sprouting angiogenesis.** First column shows the shape of vessel at time zero. Second column shows the vessel shape after four hours. Third column shows the vascular and avascular agents at 4h. Fourth column shows the results of rigid registration of the model output on the real final shape. A) Model output for a simple shape. B) Model output for a larger region. C) Model output for a region with sprouting angiogenesis (blue arrows). The parameters used for this model: $\alpha = 100$, $\beta = 40$, $\zeta = 0$, $\omega = 20$, $\eta = 0.1$, $\kappa = 20$, $\beta_{av} = 100$, $\omega_{av} = 10$, and $\kappa_{av} = 10$. Scale bar = 100 $\mu m$.

### Importance of elongation and avascular growth

Although in our model we incorporated the avascular regions' growth and vascular agents' elongation based on the *in vivo* data, we were also interested to study their relative importance in the process of remodelling. We tested this by removing the elongation and avascular growth separately and comparing their effect on remodelling in our model. Our grid search results showed us the importance of avascular presence and growth by assigning a factor of 20 to $\kappa$. Removing only the avascular growth provides us with a two phase model without the avascular growth and, although minimal, reduced the accuracy of prediction in the model (Fig 10A). Also, the model's prediction of avascular expansion into the vascular regions was impacted by removal of this growth. On the other hand, when we removed the elongation of the vascular agents we could observe a major effect of the prediction of remodelling (Fig 10B). By comparing these two results with our final model (incorporating both avascular growth and vascular agent elongation) we can see that these factors improve the prediction significantly (Fig 10C).

### Discussion

In this study, we introduced a two phase, SPP model for vascular remodelling consisting of vascular and avascular agents. The vascular agents are driven to migrate directionally, mainly

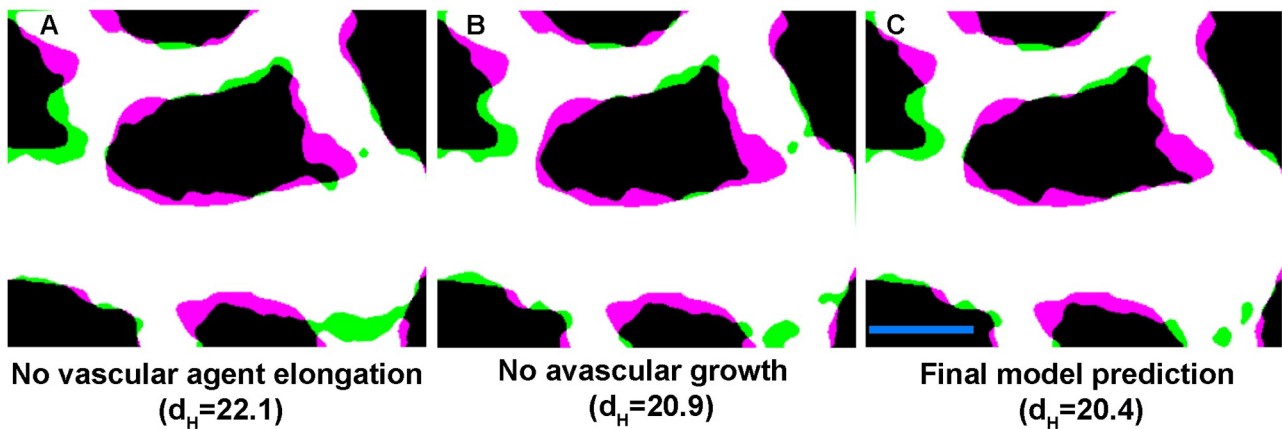

**No vascular agent elongation**
**(d$_H$=22.1)**

**No avascular growth**
**(d$_H$=20.9)**

**Final model prediction**
**(d$_H$=20.4)**

**Fig 10. Vascular agent elongation and avascular growth impact the remodelling.** A) Error in the final shape with a model without vascular agents' ellongation. B) Overlay of the final predicted shape without avascular growth. C) Overlay of the final predicted shape with the complete model. Scale bar = 100 μm.

due to shear stress. On the other hand, both the vascular and avascular agents change dimension in time which affects the remodelling process. Finally, the two types of agents interact not only with their own type (through body and remodelling forces) but also with the agents of the other type. We brought these factors together to design a model of vascular remodelling.

We constructed our model based on the hypothesis that ECs migrate in the opposite direction of flow with velocities determined by shear stress, leading to their accumulation in vessels that enlarge and migration away from vessels that regress. *In vitro*, we found that ECs have the highest migration rate at shear stress level of 0.1 Pa with the mean velocity of 10 *μm/h*. *In vivo*, and specifically during the embryonic development, different migration rates are to be expected due to changes in composition of ECM and gene expression. For this reason, we decided to use the same overall relation between shear stress and EC velocity without using the specific values found *in vitro*. Using grid search for this purpose led to assigning the highest migration velocity of 12 *μm/h* (instead of 10) to the corresponding shear stress of 0.033 Pa (instead of 0.1). Thus the optimized migration rate does not differ significantly from the experimental measured maximum, however there is a significant shift in the level of shear stress where the maximum occurs.

Our model is the first one, that we are aware of, which takes into account both vascular and avascular agents and uses interaction between the two agent types. Models by Secomb and Pries include influences from non-vascular regions in terms of changes in oxygen partial pressure, but they did not model the physical forces created by these cells as they expand [40]. The avascular region in the embryo at this stage is composed of ECM and different types of cells. Hence, changes in the surface area of these regions have an effect on vascular remodelling. Without incorporating these interactions, the model is not capable of predicting regions of ingrowth. When we removed the avascular growth, we did not see a significant effect on the similarity measurements. We believe that the contribution of avascular growth was minimal because we also scale the overall final images. Thus even with growth of avascular agents, overall growth is still incorporated in this model.

Our model is also unique in that it incorporates direction changes in shape of our agents (elliptical elongation). In our staining of the embryos, we saw that ECs elongate significantly during remodelling, in the direction of flow, while they retain the same width. This elongation is equivalent to 30% decrease in EC density. Udan et al. had previously found that no change in EC proliferation occurred during remodelling in the embryo in response to flow [7]. Our

observed changes in cell size could therefore explain how EC density decreases even without proliferation. In other words, EC elongation plays a part in vessel enlargement. Surprisingly, elongation occurred at the same rate in vessels with different diameters. We can generally assume that smaller vessels have less flow than larger vessels and yet, the rate of elongation did not differ. This means that flow induces elongation, but the rate is not proportional to shear stress levels. This does not mean that the extent of elongation (or length once equilibrium is reached) will be independent of the level of shear stress. We can only say that the kinetics of elongation is independent of vessel diameter and therefore, also likely independent of shear stress level. We then incorporated this uniform EC elongation in our model. Our initially circular agents became ellipses with a major axis in the direction of flow. This is not overly apparent in our agents (Fig 8), however, we only modelled a relatively short period of time. Our model for attraction/repulsion interaction of elliptical agents was based on both centre-centre distances and foci-centre distances of the agents, rather than just the absolute centres of the agents, as by Grégorie et al. [26].

We did not extend the model beyond four hours; because, after this period, the shear stress levels in the vasculature change significantly. This is due to the change in the geometry of the vessels together with continued heart development and strengthening of cardiac function. To extend the model further, the predicted shape of the vasculature can be extracted from the model and used as the vessel shape for a new CFD analysis. Velocity vectors for this analysis can be acquired form the time-lapse imaging of the microspheres and their $\mu$PIV output. This will provide us with new input for the model for the next four hours.

Due to multiplicity of the factors playing a role in vascular remodelling, relationship between these factors cannot be studied in a stand-alone condition. For this reason, we used the grid search method to optimise related parameters. We divided these factors into the mentioned three groups. Both shear stress and its gradient have been proposed to play a role in remodelling. From our optimisation, we found that, although shear stress plays the major role in vascular remodelling (with the highest scaling factor of all components), the shear stress gradient has no significant effect on vascular remodelling (Table 1). Our results indicate that ECs are migrating with a rate proportional to the amount of shear stress but do not follow shear stress gradients per se. However, our vascular agents do migrate as a sheet. So, if one agent feels higher shear stress levels, then the input from its neighbours will dampen its velocity. As such, it is possible that the gradients are inherent in our model and an additional parameter was not needed.

It should be pointed out that here, we did the grid search only for the vascular parameters and not for the avascular ones which are not optimised. For the avascular agents, as there is no directional migration present, its influence was removed from their model. We assigned the main importance to their body force with a small importance for the remodelling force and the body force caused by the vascular agents on the avascular agents. This was done due to the nature of avascular regions in our model as areas which are only growing in size. A more comprehensive optimisation would have involved a complete grid search for all of these parameters as well. However, our preliminary investigation of the relative weight of the factors did not lead to a significant improvement in the model and was not pursued.

We tested two more complicated vessel shapes. As long as no angiogenesis was present, our model predicted the final vessel shape fairly accurately. We previously found that flow dynamics could predict the location of sprouts [41]. We have not included that in this model because, though we could predict the sprout location along the circumference of an avascular region, we could not predict which avascular regions would sprout. This was not purely based on avascular region's size but had other unknown factors. As such, we have not been able to factor sprout formation into our prediction.

Our model approximates a three-dimensional space as two dimensional. The flow in embryonic vessels is radially symmetric, and therefore this approximation is unlikely to strongly affect the CFD analysis. For the SPP models, however, the addition of a third dimension would not only improve the model prediction but also provide a better understanding of the remodelling process. The addition would result in a more accurate modelling of the border between vascular and avascular regions and a more refined solid-fluid boundary condition. Three dimensional SPP models have previously been developed [42] and our current model can provide the basis for such an extension. The addition of new constraints for migration of vascular agents into the fluid filled part of the vessel and their interaction with avascular agents would however be necessary for this extension.

Overall, we present one of the few available computational models to predict vascular remodelling and show that we can improve on previous models by including EC elongation and growth of avascular tissue. We include both experimental and computational data to produce our model. This model has the potential to be further advanced with the addition of a model for angiogenesis. Finally, this model can be adjusted for use in other cases where remodelling occurs such as atherosclerosis and tumour vascular remodelling.

## Materials and methods

### Flow chamber

We designed a flow chamber to provide the cells with a platform for unlimited migration under laminar shear stress (Fig 2A). The flow chamber is a closed rectangular loop with length of 145 $mm$ and cross section of $6 \times 0.8$ $mm$. Between the input and output of the media, the cross section is $6 \times 0.15$ $mm$. The flow pattern and shear stress levels across the chamber were studied at varying flow rates using CFD analysis (Fig 2B).

### Cell culture

Passage 5 Human Umbilical Vein ECs (HUVECS) (Promocell, C-12208) were used for the *in vitro* experiments. The bottom surface of the flow chamber was coated with 15 $\mu g/ml$ fibronectin (Corning, 354008) for two hours at 37˚C. 15% of the cells were dyed with CellTracker Green CMFDA Dye (ThermoFisher, C2925). The mixture of dyed and non-dyed cells was seeded on the surface at density of 1600 $cells/mm^2$. The cells were cultured for 30 hours in EGM-2MV media (Promocell, C-22121) at 37˚C to attach and become fully confluent. The cells were starved for 11 hours prior to imaging (EGM-2MV with only 0.5% instead of 5% FBS and without VEGF and FGF). The cells were imaged once every 30 minutes for 5 hours. After acquisition, spots creation and tracking algorithm of Imaris 9.1 (Bitplane) software was used to track the dyed cells.

### Analysis of mouse embryo development

All experiments were performed according to the European Directive on the care and use of experimental animals (2010/63/EU) and approved by the Animal Care and Use Committee of KU Leuven (141/2016, 166/2016). CD-1 mice were mated and the presence of a vaginal plug in the morning was taken as embryonic day (E) 0.5. Embryos were dissected between embryonic day E8.5 and 9.5 and somites were counted. Yolk sacs of embryos were collected and fixed in ice cold Dent's fixative overnight. Fixed yolk sacs were then stained using anti-VE-cadherin (R&D systems, AF1002).

## Quail *in vivo* flow analysis

Simultaneous imaging of vascular remodelling and blood flow patterns was performed in quail embryos, as previously described [29]. Briefly, a pulled quartz needle and a pico-spritzer were used to inject quail embryos with acetylated LDL (AlexaFluor 488- AcLDL) and PEGylated FluoSpheres (ThermoFisher-F8803) intravascularly to label the ECs and track blood flow patterns, respectively. Embryos were placed on an upright fluorescent microscope and their vascular network was imaged every 15 minutes using Axiocam MRC camera. At the initial time-point, the motion of the fluorescent microspheres was imaged at 250 frames per second for two entire cardiac cycles (approximately two seconds) using a high-speed camera (Photron FASTCAM Ultima APX-RS). To extract the direction and magnitude of motion (velocity vectors) from the high-speed images of the microspheres, $\mu$PIV was performed using PIVlab in MATLAB (R2018b, Mathworks) [43]. $\mu$PIV results were used as input for the CFD analysis using COMSOL Multiphysics 5.4 software.

## A mathematical model of remodelling

The vessel shape and blood flow parameters were imported into MATLAB from the CFD analysis of the quail embryos described above. The vascular and avascular sections were identified from the images of the vasculature. Each of these sections were further divided into smaller constructing agents (circular with diameter $r_e = 7 \, \mu m$) with equal distances from each other. These agents were designed to have a collective migration (described below). The relation between shear stress and the migration velocity of HUVECs, calculated using the *in vitro* experiments, was used as the bases for determining agent velocities. Each agent was assigned a main migration direction ($\vec{v}_j$) in the opposite direction of flow [9].

The model was constructed based on the SPP model developed by Grégoire et al., in which, collective migration of the cells and their alignment with each other (originally proposed by Vicsek et al.) is improved by addition of a Lennard–Jones-type body force (attraction/repulsion) [23, 26]. This body force (indicated by $\vec{f}_{ij}$) ensures that the agents keep an optimal distance ($r_e$) from each other and prevents the agents from getting too close to each other. This force accounts for both flexibility of the cells and the limit that they can be squeezed. The direction of migration for each cell in the model proposed by Grégoire et al. is calculated using Eq (1):

$$\theta_i^{t+1} = arg[\alpha \sum_{j \sim i} \vec{v}_j^t + \beta \sum_{j \sim i} \vec{f}_{ij}] + \nu \xi_i^t, \tag{1}$$

where $\alpha$, $\beta$, and $\nu$ represent the importance of $\vec{v}_j$ (velocity vector—in vascular remodelling, main direction of migration), $\vec{f}_{ij}$ (body force), and $\xi_i$ (white noise error with range: $[-\pi, \pi]$) respectively in the model. In this model, the body force is calculated using Eq (2):

$$\vec{f}_{ij} = \vec{e}_{ij} \begin{cases} -\infty, & if \ r_{ij} < r_c \\ \dfrac{1}{4} \dfrac{r_{ij} - r_e}{r_a - r_e}, & if \ r_c < r_{ij} < r_a \ , \\ 1, & if \ r_a < r_{ij} < r_o \end{cases} \tag{2}$$

where $r_c = 5$ is the repulsion distance of the agents, $r_e = 7$ is the equilibrium distance (normal diameter of agents), $r_a = 9$ is the maximum diameter of the agents (the limit that they can be

enlarged), $r_o = 11$ is the maximum effective distance between the agents where agents have an attraction effect on each other, and $\vec{e}_{ij}$ is the unit vector going from $i$ to $j$.

This model was extended to include the effect of shear stress on migration by assigning a migration velocity to each agent, determined based on the calculated shear stress. The position of each agent in the next time-step is calculated in Eq (3) based on its initial position ($X_i$), velocity ($V_i$), and direction of migration $\theta_i$:

$$X_i^{t+1} = X_i^t + V_i^t \theta_i^t \Delta t, \tag{3}$$

where $\Delta t$ is the time-step. In our model, $\Delta t = 1/12$ which is equal to 5 minutes.

The growth in avascular area in each time-step was modelled in three steps. First, centre of mass of each cluster of avascular agents was calculated. Afterwards, each agent's distance from this centre was uniformly increased with a growth rate of $\Delta_{av} = 0.042$ per time-step (1/12th of hour) equivalent to a 78.5% growth in avascular area in 8 hours. Finally, the avascular agents' diameter was increased by the same rate and their body force interaction with each other ($\vec{f}_{kk'}$) and with the vascular agents ($\vec{f}'_{ki}$) was used as the determining force to provide an equilibrium distance. Hence, in Eq (1)$\alpha$ was put to zero (as the avascular agents have no directional motion) while the diameter of these agents increased in time ($r_{c_{av}}, r_{e_{av}}, r_{a_{av}}, r_{o_{av}}$). Eq (1) was rewritten as Eq (4)with the addition of two new arguments: $\vec{f}'_{ki}$ representing the force between vascular and avascular agents (calculated using Eq (2) and $\vec{f}''_{kk'}$ as the remodelling force caused by agent closeness (defined below).

$$\theta_{i_{av}}^{t+1} = arg[\beta_{av}\sum_{k \sim k'}\vec{f}_{kk'} + \kappa_{av}\sum_{i \sim k}\vec{f}'_{ki} + \omega_{av}\sum\vec{f}''_{kk'}] + v\xi_i^t, \tag{4}$$

where $\beta_{av}$ and $\kappa_{av}$ represent the importance of avascular agent's body force caused by its neighbouring avascular agents and vascular agents respectively. $\omega_{av}$ determines the importance of remodelling force in avascular area.

After the addition of avascular agents, the model is redefined as a two-phase system in which agents from each phase interact with each other and migrate in tandem. To take these additions in calculating $\theta_i$ into account, Eq (2) was rewritten by adding three extra arguments as Eq (5):

$$\theta_i^{t+1} = arg[\alpha\sum_{j \sim i}\vec{v}_j^t + \beta\sum_{j \sim i}\vec{f}_{ij} + \kappa\sum_{k \sim i}\vec{f}'_{ik} + \zeta\sum_{j \sim i}\delta\tau_{ij} + \omega\sum\vec{f}''_{ij}] + v\xi_i^t. \tag{5}$$

In this equation, $\vec{f}'_{ik}$ (calculated based on equation Eq (2)) represents the force caused by avascular cells on each vascular agent and $\kappa$ represents the importance of this force. $\omega$ represents the importance of remodelling force $\vec{f}''_{ij}$. And the importance of shear stress gradient vector ($\vec{\delta\tau}_{ij} = \tau_j - \tau_i$) between the agent and its nearby agents is represented by $\zeta$.

The remodelling force ($\vec{f}''_{ij}$) is calculated using Eq (6). If the distance between two agents is less than $r_e$ the direction of force on it will be towards outside (nearest vessel wall, $\vec{e}_{\theta_{out}}$) and if this distance is more than $r_e$ and inside the area of interaction ($r_o$) the direction of force will be inwards (towards the vessel centre line for vascular and away from the nearest vessel walls for

avascular agents, $\vec{e}_{\theta_{in}}$):

$$\vec{f}''_{ij} = \begin{cases} r_{ij} - r_e \vec{e}_{\theta_{out}}, & if \ r_{ij} < r_e \\ r_{ij} - r_e \vec{e}_{\theta_{in}}, & if \ r_e < r_{ij} < r_o \end{cases}. \tag{6}$$

The elongation of agents throughout the remodelling process was taken into account in calculation of $\vec{f}_{ij}$. In each step, the agents were assumed to be aligned to and elongated in the direction of flow. The body force between two adjacent agents (in Fig 3, S and S' represent centres of agents $i$ and $j$ respectively) was set as the sum of three forces (Eq (7)) that were caused by the two foci (G and G') and the centre (S') of agent $j$ on centre of agent $i$ (S):

$$\vec{f}_{ij} = \vec{f}_{S'S} + \vec{f}_{GS} + \vec{f}_{G'S}. \tag{7}$$

The equilibrium distance between the two agents can be approximated using Eq (8):

$$\begin{cases} L_e = L_1 + L_2 \\ L_{e_G} = L_{G1} + L_{G2} \\ L_{e'_G} = L_{G'1} + L_{G'2} \end{cases}, \tag{8}$$

where $L_e$, $L_{e_G}$, and $L_{e'_G}$ are the equilibrium distances between S-S', S-G, and S-G' respectively. Repulsion distances of the agents ($L_c$, $L_{c_G}$, and $L_{c'_G}$), maximum extents of the agents ($L_a$, $L_{a_G}$, and $L_{a'_G}$), and maximum effective distances between the agents ($L_o$, $L_{o_G}$, and $L_{o'_G}$) can also be calculated using the same equations.

To calculate the three forces caused by the two foci and the centre of agent $j$ on centre of agent $i$, Eq (2) can be rewritten according to the new distances. $\vec{f}_{S'S}$ is calculated using Eq (9):

$$\vec{f}_{S'S} = \vec{e}_{SS'} \begin{cases} -\infty, & if \ L < L_c \\ \dfrac{1}{4}\dfrac{L - L_e}{L_a - L_e}, & if \ L_c < L < L_a \\ 1, & if \ L_a < L < L_o \end{cases}, \tag{9}$$

where $L$ is the distance between S and S' and $\vec{e}_{SS'}$ is the unit vector from S to S'. To calculate $\vec{f}_{GS}$, in this equation, S' should be replaced by G, $L$ by $L_G$, $L_e$ by $L_{eG}$, $L_c$ by $L_{cG}$, and $L_a$ by $L_{aG}$ and to calculate $\vec{f}_{G'S}$, S' should be replaced by G', $L$ by $L'_G$, $L_e$ by $L_{eG'}$, $L_c$ by $L_{cG'}$, and $L_a$ by $L_{aG'}$.

In our model, the vascular agents initially have a circular shape with a diameter of $r_e$. In each time-step, the agents elongate with a rate of $\Delta = 0.003125$ (equivalent to 30% increase in length over eight hours) purely in the direction of flow. This means that the dimensions of the elliptical agents will be defined by semi-major axis $a = 1.3 \times (0.5 \times r_e)$ and semi-minor axis $b = 0.5 \times r_e$.

The model was tested for a time period of 4 hours. Three vessel shapes from three different embryos were selected. The final position of the agents was then used to make a binary image based on their final dimensions ($a$ and $b$) to predict the shape of the vessel. To account for the non-rigidity of ECs, a dilation and an erosion were used in sequence with a disk element of radius 2 on the final binary image of vascular pixels. Afterwards, the output image of the model was scaled to account for the overall growth and using rigid registration (translation/rotation) was overlaid on the final shape of the vessel in reality. To compare the final shape of

the vessel in reality with its prediction, Dice similarity coefficient and Hausdorff distance ($d_H$) [44] were used.

## Grid search

Grid search was performed on three different sets of parameter separately in MATLAB. For the parameters controlling shear stress driven migration the range was determined by the *in vitro* results. For the two other sets, the relative importance of each parameter in each case was set such that the range covered from zero up to 100 times the importance of the other factors. The similarity between the predictions and reality was compared in each case and the sets of values with the best prediction was selected for each parameter.

## Supporting information

**S1 Fig. Assigning the shear stress levels felt by ECs in the model.** A) Shear stress levels within the fluid and along the vessel wall. B) Calculating the shear stress level for each point throughout the vasculature from interpolation of the nearest walls' shear stress levels.
(TIF)

**S2 Fig. Characterisation of EC monolayer.** Cells were seeded, as for shear stress experiments, in our chambers and then fix and stained for VE-Cadherin. Monolayers showed continuous and straight cell-cell junctions within the endothelial layer. Nuclei were labelled with ToPro.
(TIF)

**S3 Fig. Sensitivity analysis of the optimised parameters.** Parameters were varied to identify how errors in each parameter would affect the final prediction for A) $V_{max}$ which is the maximum velocity of migration at $\tau_{max}$, B) $\tau_{max}$ which is the value of shear stress at which the maximum migration occurs, C) $\alpha$ which is the importance of velocity vectors in alignment of adjacent agents, D) $\zeta$ which is the importance of shear stress gradients, E) $\beta$ which is the importance of body forces between vascular agents, F) $\kappa$ which is the importance of body forces from avascular agents on vascular agents, and G) $\omega$ which is the remodelling force.
(TIF)

**S1 File. Full results of the grid search analysis for the maximum agent velocity and its corresponding shear stress.**
(XLSX)

**S2 File. Full results of the grid search analysis for factors controlling shear stress induced migration and shear stress gradient.**
(XLSX)

**S3 File. Full results of the grid search analysis for the inter-agent forces.**
(XLSX)

**S1 Video. AF488-AcLDL time-lapse of vascular remodelling of quail embryo during development.**
(MP4)

**S2 Video. Migration of agents in the model.** The agents defined by the model migrate in each time-step while the vascular agents elongate and the diameter of avascular agents increase.
(MP4)

## Acknowledgments

We thank Hanna Peacock and Steven Simmonds for their input regarding the experimental and computational design of the model.

## Author Contributions

**Conceptualization:** Ashkan Tabibian, Diego A. Vargas, Hans Van Oosterwyck, Elizabeth A. V. Jones.

**Data curation:** Elizabeth A. V. Jones.

**Formal analysis:** Ashkan Tabibian.

**Funding acquisition:** Hans Van Oosterwyck, Elizabeth A. V. Jones.

**Investigation:** Ashkan Tabibian, Siavash Ghaffari, Diego A. Vargas.

**Methodology:** Ashkan Tabibian, Siavash Ghaffari, Diego A. Vargas, Hans Van Oosterwyck, Elizabeth A. V. Jones.

**Project administration:** Elizabeth A. V. Jones.

**Resources:** Elizabeth A. V. Jones.

**Software:** Ashkan Tabibian, Diego A. Vargas.

**Supervision:** Elizabeth A. V. Jones.

**Validation:** Ashkan Tabibian.

**Visualization:** Ashkan Tabibian.

**Writing – original draft:** Ashkan Tabibian, Elizabeth A. V. Jones.

**Writing – review & editing:** Ashkan Tabibian, Siavash Ghaffari, Diego A. Vargas, Hans Van Oosterwyck, Elizabeth A. V. Jones.

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
