## [Decision Letter · Decision Letter 0]

22 May 2020

Dear Dr. Jones,

Thank you very much for submitting your manuscript "Simulating flow induced migration in vascular remodelling" for consideration at PLOS Computational Biology.

As with all papers reviewed by the journal, your manuscript was reviewed by members of the editorial board and by several independent reviewers. In light of the reviews (below this email), we would like to invite the resubmission of a significantly-revised version that takes into account the reviewers' comments.

Special consideration should be given to the reviewer comment relating to vascular remodeling in 3 dimensions and the limits of 2D experiemental data.

We cannot make any decision about publication until we have seen the revised manuscript and your response to the reviewers' comments. Your revised manuscript is also likely to be sent to reviewers for further evaluation.

Sincerely,

Scott L Diamond

Associate Editor

PLOS Computational Biology

Jason Haugh

Deputy Editor

PLOS Computational Biology

Special consideration should be given to the reviewer comment relating to vascular remodeling in 3 dimensions and the limits of 2D experiemental data.

Reviewer's Responses to Questions

**Comments to the Authors:**

Reviewer #1: In this manuscript, Tabibian et al. design, develop and validate a model of vascular remodeling during embryogenesis. The model is based on experimental data, collected both in vivo and in vitro to define values and ranges of flow-dependent hemodynamic conditions, cell shape and migration. The model predictions are challenged against real experimental observation and the relevance of different parameters is evaluated.

The results obtained are overall interesting, with a well-crafted set of experiments to describe the evolution of both vascular and non-vascular components, with reference to the overall development of the body. The manuscript should be accepted after the noted changes:

1. The distinction between vascular and non-vascular agents is clear from the perspective of the model, much less for the biology. Vascular agents are endothelial cells, but then what are non-vascular agents? The authors should clarify if their categorisation is purely based on the imaging, or it has a biological meaning, which then can be used in the interpretation of the results.

2. HUVEC monolayers used in flow experiments should be better characterised. I am missing some IF staining confirming that the reconstituted tissue is fully mature at the time of flow onset. VE-cadherin distribution and density evaluation should be enough.

3. It looks like some figures and quantifications do not report the number of independent experiments.

4. In my experience the cell tracker staining can affect the migration of HUVEC. Limiting the % of cells receiving the staining surely helps, but due to the relevance of these experimental data, I suggest to have a control evaluation. A reference wound healing experiment could be used to test whether in these experimental conditions collective cell migration is preserved.

Reviewer #2: This paper presents a computational model of vascular remodeling based on in-vitro and in-vivo experimental data in the embryo yolk sac. Specifically, a simple agent-based model takes into account shear stress, cell migration and EC density during embryonic development. It is well-known that shear stress is a key regulator of vascular remodeling, but the details of the mechanisms are not known. Overall, this is an interesting study with potential relevance to vascular development. However, there are a number of issues with the approach that should be addressed.

1) This model is based on a previous agent-based model and has been slightly improved by including a term for shear stress to include its effect on ECs migration. However, it is not clear from the results how this term improves the prediction of migration by ECs.

2) This model has many theoretical parameters that cannot be determined experimentally. If it is not possible to experimentally determine these parameters at least a sensitivity analysis should be performed to show the sensitivity of the results to these parameters.

3) Some experimental results (Fig. 6 at hr 4) are not in agreement with computational results. The authors should further discuss the discrepancies

4) The authors should cite the seminal works of Secomb and Pries, who pioneered the area of shear-induced vascular remodeling.

5) Collective migration is a result of cell crowding (which the authors reproduce as repulsive forces), but also cell-cell adhesion (which is not considered in the model). Would the results be improved if there were some tendency for cells to “drag” each other due to adhesive forces?

6) Most importantly, it is not clear that the presented 2D model reproduces accurately the 3D tissue being considered. In reality, cells in the avascular space have an extra degree of freedom to move into or out of the 2D domain. Similarly, cells in the vessel really don’t have a fixed boundary preventing migration laterally. They are bounded by basement membrane around the circumference, but can move with the same freedom around the circumference or along the axis of the vessel. To accommodate the reduction to 2 dimensions, the authors made some critical assumptions about the interactions between the endothelial cells and the surrounding tissue at the boundaries, which are very suspect. Thus, it is unlikely that the presented model truly reproduces the mechanisms present in vivo.

**Have all data underlying the figures and results presented in the manuscript been provided?**

Reviewer #1: Yes

Reviewer #2: Yes

PLOS authors have the option to publish the peer review history of their article (what does this mean?). If published, this will include your full peer review and any attached files.

Reviewer #1: Yes: Aldo Ferrari

Reviewer #2: No
---

## [Decision Letter · Decision Letter 1]

17 Jul 2020

Dear Dr. Jones,

We are pleased to inform you that your manuscript 'Simulating flow induced migration in vascular remodelling' has been provisionally accepted for publication in PLOS Computational Biology.

Before your manuscript can be formally accepted you will need to complete some formatting changes, which you will receive in a follow up email. A member of our team will be in touch with a set of requests. Please also note that Reviewer #2 has suggested that some discussion of model limitations related to 3D vs. 2D be added, which we encourage you to add to your final manuscript.

Best regards,

Scott L Diamond

Associate Editor

PLOS Computational Biology

Jason Haugh

Deputy Editor

PLOS Computational Biology

Reviewer's Responses to Questions

**Comments to the Authors:**

Reviewer #2: The authors have responded adequately to the previous review, except for the last comment regarding 2D vs 3D modeling. The authors need to at least include some discussion about their assumptions-- specifically regarding EC interactions with the surrounding tissue -- and how these might be different in their 2D model compared to a 3D vessel.

**Have all data underlying the figures and results presented in the manuscript been provided?**

Reviewer #2: None

PLOS authors have the option to publish the peer review history of their article (what does this mean?). If published, this will include your full peer review and any attached files.

Reviewer #2: No

---

## [Editor Report · Acceptance letter]

11 Aug 2020

PCOMPBIOL-D-20-00575R1 

Simulating flow induced migration in vascular remodelling

Dear Dr Jones,

I am pleased to inform you that your manuscript has been formally accepted for publication in PLOS Computational Biology. Your manuscript is now with our production department and you will be notified of the publication date in due course.

With kind regards,

Laura Mallard
